# Entropy Generation and Mixed Convection Flow Inside a Wavy-Walled Enclosure Containing a Rotating Solid Cylinder and a Heat Source

**DOI:** 10.3390/e22060606

**Published:** 2020-05-29

**Authors:** Ammar I. Alsabery, Tahar Tayebi, Rozaini Roslan, Ali J. Chamkha, Ishak Hashim

**Affiliations:** 1Refrigeration & Air-conditioning Technical Engineering Department, College of Technical Engineering, The Islamic University, 54001 Najaf, Iraq; alsabery_a@iunajaf.edu.iq; 2Department of Mathematical Sciences, Faculty of Science & Technology, Universiti Kebangsaan Malaysia, UKM Bangi 43600, Selangor, Malaysia; 3Faculty of Sciences & Technology, Mohamed El Bachir El Ibrahimi University, Bordj Bou Arreridj, El-Anasser, Algeria; tahartayebi@gmail.com; 4Energy Physics Laboratory, Department of Physics, Faculty of Science, Mentouri Brothers Constantine 1 University, Constantine, Algeria; 5Department of Mathematics & Statistics, Faculty of Applied Sciences & Technology, Universiti Tun Hussein Onn Malaysia, Pagoh, Muar 84600, Johor, Malaysia; rozaini@uthm.edu.my; 6Institute of Research and Development, Duy Tan University, Da Nang 550000, Vietnam; 7Institute of Theoretical and Applied Research (ITAR), Duy Tan University, Hanoi 100000, Vietnam

**Keywords:** rotating inner cylinder, heat source, entropy generation, mixed convection, heat transfer, wavy cavity

## Abstract

The current study investigates the 2D entropy production and the mixed convection inside a wavy-walled chamber containing a rotating cylinder and a heat source. The heat source of finite-length *h* is placed in the middle of the left vertical surface in which its temperature is fixed at $T_{h}$. The temperature of the right vertical surface, however, is maintained at lower temperature $T_{c}$. The remaining parts of the left surface and the wavy horizontal surfaces are perfectly insulated. The governing equations and the complex boundary conditions are non-dimensionalized and solved using the weighted residual finite element method, in particular, the Galerkin method. Various active parameters are considered, i.e., Rayleigh number $Ra=10^{3}$ and $10^{5}$, number of oscillations: 1≤N≤4, angular rotational velocity: −1000≤Ω≤1000, and heat source length: 0.2≤H≤0.8. A mesh independence test is carried out and the result is validated against the benchmark solution. Results such as stream function, isotherms and entropy lines are plotted and we found that fluid flow can be controlled by manipulating the rotating velocity of the circular cylinder. For all the considered oscillation numbers, the Bejan number is the highest for the case involving a nearly stationary inner cylinder.

## 1. Introduction

The mechanisms of natural and mixed convection in closed cavities of various shapes such as triangular, rectangular, square, elliptical, cylindrical, and spherical have been extensively studied. Some complex geometrical configurations such as wavy, convex and concave curved wall cavities, etc. have been studied as well. The applications of convective heat transfer in complex geometries can be found in many engineering problems such as electrical and nuclear components, micro-electronic devices, solar collectors, etc. [1]. The convective heat transfer and fluid flow in a vertical wavy enclosure filled with a porous medium were examined numerically by Kumar [2]. It was observed that the heat transfer was significantly affected by wave phase, wave amplitude, number of waves and Rayleigh number. Adjlout et al. [3] investigated the laminar free convection inside a tilted hollow with a wavy surface. Their numerical outcomes showed that fluid flow and heat transfer rate in the cavity were affected by the heated wavy surface. Das and Mahmud [4] analyzed the natural convection in a cavity with two isothermal horizontal curved walls. The vertical walls were treated as adiabatic. They found that the heat transfer characteristics in the cavity were influenced by the number of undulations and the amplitude of the wavy wall. Al-Amiri et al. [5] examined numerically the mixed convection inside a lid-driven enclosure with a curved base surface. The undulating wall was heated sinusoidally, while the two vertical surfaces were insulated. The flow and heat transfer features at different Richardson numbers, amplitudes of wavy surface and undulation numbers were studied. Dalal and Das [6] investigated the convection within a cavity, including a curved vertical surface. They found that the flow behavior and the heat transfer rate were influenced by the presence of undulation in the wavy boundary. Rostami [7] investigated the unsteady heat transfer within a cavity with straight horizontal surfaces and vertical undulating surfaces. Flow parameters (i.e., Grashof number, Prandtl number) and geometrical parameters (aspect ratio = average width/wavelength and wave ratio = amplitude/wavelength) were analyzed. Abu-Nada and Chamkha [8] studied the combined convection inside a lid-driven cavity filled with a water–CuO nanofluid. The lid is heated and controlled by a fixed hot temperature whereas the wavy bottom wall was set at lower temperature conditions. The left and the right boundaries of the enclosure were insulated. For all Richardson numbers, Abu-Nada and Chamkha [8] showed that the heat transfer rate was heavily dependent on the nanoparticles volume fraction and the wavy wall geometry ratios. Sheikholeslami and Chamkha [9] simulated the flow and convective heat transfer inside a lid-driven cavity with a curved wall subjected to a magnetic field of varying strength.

The thermo-fluid study involving rotating elements has been gaining popularity nowadays due to its practicability in engineering applications such as drilling in oil wells, turbomachinery, rotating-tube heat exchangers, chemical mixing devices and fuel rod in nuclear reactors. Hayase et al. [10] investigated the heat transfer between the periodically embedded cavities with rotating coaxial cylinders. They found that heat transfer and momentum increased by factors of 1.1–1.2 if the cavities were embedded within an exterior or inner cylinder, respectively. Fu et al. [11] adopted the finite-element method to investigate free convection inside a cavity with an inner rotating cylinder. Ghaddar [12] simulated the mixed convective heat transfer in a rectangular cavity containing a rotating cylindrical heat source. The author reported that the circumrotation of the cylinder could intensify the heat transfer rate and the degree of enhancement, depending on the location of the cylinder. Mixed convection in the rectangular cavity containing a rotating plate was studied experimentally by Kimura et al. [13]. Oztop et al. [14] used statistical research to study the combined convective heat transfer and fluid flow in a lid-driven square cavity containing a circular body. They found that the circular body would change the heat and fluid flow patterns significantly. Paramane and Sharma [15] simulated the two-dimensional forced convection over the uniformly heated rotating cylinder. Their results proved that heat transfer can be suppressed by controlling the rotational motion of the circular cylinder. Liao and Lin [16] utilized the immersed-boundary method to simulate the free and mixed convections inside a square cavity containing an active rotating circular cylinder. They presumed that the movement of the cylinder could lower the heat transfer rate. Hussain and Hussein [17] examined the mixed convection numerically in a square cavity filled with air. The cavity contained a conductive rotating cylinder. Chatterjee et al. [18] simulated the mixed convection inside the lid-driven cavity filled with nanofluid. The cavity contained a thermally insulated rotating cylinder in the core. The authors confirmed the dependence of the heat transfer rate on the the rotational rate of the cylinder. The mixed convection inside a square cavity including a rotating cylinder was studied by Costa and Raimundo [19]. They studied the effects of radius, crossing velocity, thermal conductivity and thermal capacity of the rotating cylinder on the overall thermal performance. Khanafer and Aithal [20] applied the commercial software ADINA to simulate the mixed convection inside a cavity containing a rotating cylinder. The effects of Richardson number, rotating velocity and rotational direction were analyzed. Their result showed that the average Nusselt number increased with respect to the rotational speed. Heat and mass transfers generate irreversibility (entropy generation), thus leading to a loss in the efficiency of a real process. Many research works related to entropy generation in cavities can be found in open literature (see Mahmud and Island [21], Magherbi et al. [22], Zahmatkesh [23], Voral et al. [24], Ilis et al. [25], Parvin and Chamkha [26], Chamkha et al. [27], Sheremet et al. [28], Mansour et al. [29], Chamkha et al. [30] and Mehryan et al. [31]).

To the best of our knowledge, the entropy generation and mixed convection inside a cavity with horizontal wavy walls containing a solid rotating circular cylinder has not been studied yet. Therefore, in the current work, the effects of the length of the heater, the amplitude, the angular velocity direction and the number of oscillations of the wavy walls on the thermo-hydrodynamic behavior and the entropy generation inside the cavity is investigated. The two horizontal wavy walls bounding the cavity are treated as adiabatic. This problem can be found in many practical heat transfer applications such as thermal managements in electric machinery and electronic components, oil and water extraction, enhanced oil displacement, friction reduction in lubricants in engineered systems, etc. [32,33,34,35,36].

## 2. Mathematical Formulation

The steady, two-dimensional free convection in a wavy chamber of length *L* is studied. The chamber contains a rotating circular cylinder of thermal conductivity *k*, which is placed in the center as shown in Figure 1. The right vertical surface of the cavity is maintained at a cold temperature $T_{c}$. Meanwhile, a heater of length *h* and maintained at hot temperature $T_{h}$ is placed within the left vertical surface. The remaining parts of the left surface and the wavy horizontal walls are perfectly insulated. The edges of the flow domain are impermeable, and the region between the surfaces of the chamber and the solid cylinder is filled with water (Pr=4.623). The Boussinesq approximation is adopted. Therefore, the continuity, momentum and energy equations of the laminar Newtonian fluid flow are [37]: (1)$$\begin{array}{c} \frac{\partial u}{\partial x}+\frac{\partial v}{\partial y}=0,\\ u\frac{\partial u}{\partial x}+v\frac{\partial u}{\partial y}=-\frac{1}{\rho_{f}}\frac{\partial p}{\partial x}+\nu_{f}\;\left(\frac{\partial^{2}u}{\partial x^{2}}+\frac{\partial^{2}u}{\partial y^{2}}\right), \end{array}$$(2)$$\begin{array}{c} u\frac{\partial v}{\partial x}+v\frac{\partial v}{\partial y}=-\frac{1}{\rho_{f}}\frac{\partial p}{\partial y}+\nu_{f}\;\left(\frac{\partial^{2}v}{\partial x^{2}}+\frac{\partial^{2}v}{\partial y^{2}}\right)+\beta_{f}g\left(T-T_{0}\right), \end{array}$$(3)$$\begin{array}{c} u\frac{\partial T}{\partial x}+v\frac{\partial T}{\partial y}=\alpha_{f}\left(\frac{\partial^{2}T}{\partial x^{2}}+\frac{\partial^{2}T}{\partial y^{2}}\right). \end{array}$$

As the inner cylinder is modelled as a moving mass block (via an external force), the energy equation of the solid cylinder is:(4)$$u_{s}\frac{\partial T}{\partial x}+v_{s}\frac{\partial T}{\partial y}=\frac{k_{s}}{{\left(\rho C_{p}\right)}_{s}}\left(\frac{\partial^{2}T}{\partial x^{2}}+\frac{\partial^{2}T}{\partial y^{2}}\right).$$

Here, *x* and *y* are the horizontal and vertical directions, *u* and *v* are the velocity components in the *x*- and *y*-directions, respectively, and $|V_{s}|=\sqrt{u_{s}^{2}+v_{s}^{2}}=r\times \left|\omega \right|$ is the vector velocity of the solid cylinder surface. $\nu_{f}$ denotes the kinematic viscosity of the liquid, $\beta_{f}$ is the thermal expansion coefficient of the liquid, *g* is the gravitational acceleration, *T* is the temperature and $\alpha_{f}=\frac{k_{f}}{{\left(\rho C_{p}\right)}_{f}}$ is the thermal diffusivity of the fluid.

The dimensionless variables are:(5)$$\begin{array}{c} X=\frac{x}{L},\;Y=\frac{y}{L},\;U=\frac{uL}{\alpha_{f}},\;V=\frac{vL}{\alpha_{f}},\;\theta =\frac{T-T_{c}}{T_{h}-T_{c}},\;\theta_{s}=\frac{T_{s}-T_{c}}{T_{h}-T_{c}},\\ \;R=\frac{r}{L},\mathrm{Pr}=\frac{\nu_{f}}{\alpha_{f}},\;Ra=\frac{g\beta_{f}\left(T_{h}-T_{c}\right)L^{3}}{\nu_{f}\alpha_{f}},\;P=\frac{pL^{2}}{\rho_{f}\alpha_{f}^{2}},\;\Omega =\frac{\omega L^{2}}{\alpha_{f}}, \end{array}$$
and the resulting non-dimensional governing equations are: (6)$$\begin{array}{c} \frac{\partial U}{\partial X}+\frac{\partial V}{\partial Y}=0, \end{array}$$(7)$$\begin{array}{c} U\frac{\partial U}{\partial X}+V\frac{\partial U}{\partial Y}=-\frac{\partial P}{\partial X}+\mathrm{Pr}\;\left(\frac{\partial^{2}U}{\partial x^{2}}+\frac{\partial^{2}U}{\partial Y^{2}}\right), \end{array}$$(8)$$\begin{array}{c} U\frac{\partial V}{\partial X}+V\frac{\partial V}{\partial Y}=-\frac{\partial P}{\partial Y}+\mathrm{Pr}\;\left(\frac{\partial^{2}V}{\partial X^{2}}+\frac{\partial^{2}V}{\partial Y^{2}}\right)+Ra\mathrm{Pr}\theta , \end{array}$$(9)$$\begin{array}{c} U\frac{\partial \theta }{\partial X}+V\frac{\partial \theta }{\partial Y}=\frac{\partial^{2}\theta }{\partial X^{2}}+\frac{\partial^{2}\theta }{\partial Y^{2}}, \end{array}$$(10)$$\begin{array}{c} U_{s}\frac{\partial \theta }{\partial X}+V_{s}\frac{\partial \theta }{\partial Y}=\frac{{\left(\rho C_{p}\right)}_{f}}{{\left(\rho C_{p}\right)}_{s}}\frac{k_{s}}{k_{f}}\left(\frac{\partial^{2}\theta }{\partial X^{2}}+\frac{\partial^{2}\theta }{\partial Y^{2}}\right). \end{array}$$

Eqsuations (Equation 6)–(Equation 10) need to be solved subject to the dimensionless boundary conditions as follows: $$\begin{array}{c} \mathrm{At}\mathrm{the}\mathrm{heated}\mathrm{segment}\mathrm{of}\mathrm{the}\mathrm{left}\mathrm{vertical}\mathrm{surface}: \end{array}$$(11)$$\begin{array}{c} U=V=0,\;\;\theta =1:\;\;X=0,\;\;(1-H)/2\leq Y\leq (1+H)/2.\\ \mathrm{At}\mathrm{the}\mathrm{adiabatic}\mathrm{portions}\mathrm{of}\mathrm{the}\mathrm{left}\mathrm{surface}: \end{array}$$(12)$$\begin{array}{c} U=V=0,\;\;\frac{\partial \theta }{\partial X}=0:\;\;X=0,\;\;0\leq Y\leq \left(1-H\right)/2\;\;\mathrm{and}\;\;\left(1+H\right)/2\leq Y\leq 1.\;\;\\ \mathrm{At}\mathrm{the}\mathrm{right}\mathrm{vertical}\mathrm{surface}: \end{array}$$(13)$$\begin{array}{c} U=V=0,\;\;\theta =0:\;\;X=1,\;\;0\leq Y\leq 1.\\ \mathrm{At}\mathrm{the}\mathrm{adiabatic}\mathrm{top}\mathrm{wavy}\mathrm{surface}: \end{array}$$(14)$$\begin{array}{c} U=V=0,\;\;\frac{\partial \theta }{\partial n}=0:\;\;1-A\left(1-\cos\left(2N\pi \times \left[0\leq Y\leq 1\right]\right)\right),\;\;0\leq X\leq 1.\\ \mathrm{At}\mathrm{the}\mathrm{adiabatic}\mathrm{bottom}\mathrm{wavy}\mathrm{wall}: \end{array}$$(15)$$\begin{array}{c} U=V=0,\;\;\frac{\partial \theta }{\partial n}=0:\;\;A\left(1-\cos\left(2N\pi \times \left[0\leq Y\leq 1\right]\right)\right),\;\;0\leq X\leq 1.\\ \mathrm{At}\mathrm{the}\mathrm{outside}\mathrm{surface}\mathrm{of}\mathrm{the}\mathrm{solid}\mathrm{cylinder}: \end{array}$$(16)$$\begin{array}{c} U=-\Omega \left(Y-Y_{0}\right),\;V=\Omega \left(X-X_{0}\right),\;\;\theta =\theta_{s},\;\;\frac{\partial \theta }{\partial n}=K_{r}\frac{\partial \theta_{s}}{\partial n}. \end{array}$$

$Ri=\left(Ra/\mathrm{Pr}\right)/Re^{2}$ is the modified Richardson number introduced to determine the effects of natural and forced convections, where Re is the Reynolds number. However, in the current work, Ri is defined as [19]:(17)$$Ri=\frac{Ra\cdot \mathrm{Pr}}{4\Omega^{2}\cdot R^{4}},$$
for Ω≠0 and R≠0. The local Nusselt number of the heated portion of the left vertical surface is expressed as:(18)$$Nu=-{\left(\frac{\partial \theta }{\partial X}\right)}_{X=0}.$$

The local Nusselt number at the interface of the solid cylinder is defined as [19]:(19)$$Nu_{i}=-\frac{k_{s}}{k_{f}}{\left(\frac{\partial \theta }{\partial n}\right)}_{r=R}.$$

Lastly, the average Nusselt number of the heated portion of the left vertical surface is expressed as:(20)$$\begin{array}{c} \bar{Nu}=\int_{\frac{1-H}{2}}^{\frac{1+H}{2}}Nu\;dX, \end{array}$$
while the average Nusselt number at the interface is:(21)$$\begin{array}{c} {\bar{Nu}}_{i}=\int_{0}^{\Theta }Nu_{i}\;dn. \end{array}$$

The entropy generation is defined as [25,37,38]:(22)$$\begin{array}{c} S=\frac{k_{f}}{T_{0}^{2}}\left[{\left(\frac{\partial T}{\partial x}\right)}^{2}+{\left(\frac{\partial T}{\partial y}\right)}^{2}\right]+\frac{\mu_{f}}{T_{0}}\left[2{\left(\frac{\partial u}{\partial x}\right)}^{2}+2{\left(\frac{\partial v}{\partial y}\right)}^{2}+{\left(\frac{\partial u}{\partial x}+\frac{\partial v}{\partial x}\right)}^{2}\right]. \end{array}$$

The dimensionless local entropy production rate is:(23)$$\begin{array}{c} S_{\mathrm{GEN}}={\left(\frac{\partial \theta }{\partial X}\right)}^{2}+{\left(\frac{\partial \theta }{\partial Y}\right)}^{2}+\phi \left\{2\left[{\left(\frac{\partial U}{\partial X}\right)}^{2}+{\left(\frac{\partial V}{\partial Y}\right)}^{2}\right]+{\left(\frac{\partial^{2}U}{\partial Y^{2}}+\frac{\partial^{2}V}{\partial X^{2}}\right)}^{2}\right\}, \end{array}$$
where $S_{\mathrm{GEN}}=S_{\mathrm{gen}}\frac{T_{0}^{2}L^{2}}{k_{f}{\left(\Delta T\right)}^{2}}$ and $\phi =\frac{\mu_{f}T_{0}}{k_{f}}{\left(\frac{\alpha_{f}}{L(\Delta T)}\right)}^{2}$ is the irreversibility distribution rate. Equation (Equation 23) can be written as:(24)$$\begin{array}{c} S_{\mathrm{GEN}}=S_{\theta }+S_{\Psi }, \end{array}$$
where $S_{\theta }$ is the entropy generation due to heat transfer irreversibility (HTI) and $S_{\Psi }$ is entropy generation due to fluid friction irreversibility (FFI):(25)$$\begin{array}{cc} & S_{\theta }={\left(\frac{\partial \theta }{\partial X}\right)}^{2}+{\left(\frac{\partial \theta }{\partial Y}\right)}^{2}, \end{array}$$(26)$$\begin{array}{c} S_{\Psi }=\phi \left\{2\left[{\left(\frac{\partial U}{\partial X}\right)}^{2}+{\left(\frac{\partial V}{\partial Y}\right)}^{2}\right]+{\left(\frac{\partial^{2}U}{\partial Y^{2}}+\frac{\partial^{2}V}{\partial X^{2}}\right)}^{2}\right\}. \end{array}$$

The global entropy generation (GEG) can be computed by integrating Equation (Equation 24) over the computational region:(27)$$\begin{array}{c} \mathrm{GEG}=\int S_{\theta }\mathrm{dXdY}+\int S_{\Psi }\mathrm{dXdY}. \end{array}$$

The Bejan number can be used to check the relative dominance of heat transfer and fluid friction irreversibility. It can be calculated as:(28)$$\begin{array}{c} Be=\frac{\int S_{\theta }\;dXdY}{\int S_{\mathrm{GEN}}\;dXdY}. \end{array}$$

Accordingly, Be>0.5 indicates that irreversibility is dominated by HTI.

## 3. Numerical Method and Validation

The dimensionless governing Equations (Equation 6)–(Equation 10) subjected to the boundary conditions (Equation 11)–(Equation 16) are solved using the weighted residual finite element method (i.e., the Galerkin method). The finite element procedure for solving Equations (Equation 7) and (Equation 8) is given by applying the following scheme.

First, the penalty finite element method is utilized by excluding the pressure (*P*) and including a penalty parameter (λ) in the following manner:$$\begin{array}{c} P=-\lambda \left(\frac{\partial U}{\partial X}+\frac{\partial V}{\partial Y}\right). \end{array}$$

The following equations in the *X* and *Y*-directions can then be formulated as:$$\begin{array}{c} U\frac{\partial U}{\partial X}+V\frac{\partial U}{\partial Y}=\frac{\partial \lambda }{\partial X}\left(\frac{\partial U}{\partial X}+\frac{\partial V}{\partial Y}\right)+\mathrm{Pr}\left(\frac{\partial^{2}U}{\partial X^{2}}+\frac{\partial^{2}U}{\partial Y^{2}}\right),\\ U\frac{\partial V}{\partial X}+V\frac{\partial V}{\partial Y}=\frac{\partial \lambda }{\partial Y}\left(\frac{\partial U}{\partial X}+\frac{\partial V}{\partial Y}\right)+\mathrm{Pr}\left(\frac{\partial^{2}V}{\partial X^{2}}+\frac{\partial^{2}V}{\partial Y^{2}}\right)+Ra\mathrm{Pr}\theta . \end{array}$$

The weighted-integral (or weak) formulation of the momentum equations can be expressed by integrating the above formulation over the computational area consisting of triangular elements as shown in Figure 2. The next weak formulation can be expressed as: $$\begin{array}{c} \int_{\Omega }\left(\Phi_{i}U^{k}\frac{\partial U^{k}}{\partial X}+\Phi_{i}V^{k}\frac{\partial U^{k}}{\partial Y}\right)dXdY=\lambda \int_{\Omega }\frac{\partial \Phi_{i}}{\partial X}\left(\frac{\partial U^{k}}{\partial X}+\frac{\partial V^{k}}{\partial Y}\right)dXdY\\ \;+\mathrm{Pr}\int_{\Omega }\Phi_{i}\left(\frac{\partial^{2}U^{k}}{\partial X^{2}}+\frac{\partial^{2}U^{k}}{\partial Y^{2}}\right)dXdY,\\ \int_{\Omega }\left(\Phi_{i}V^{k}\frac{\partial V^{k}}{\partial X}+\Phi_{i}V^{k}\frac{\partial V^{k}}{\partial Y}\right)dXdY=\lambda \int_{\Omega }\frac{\partial \Phi_{i}}{\partial Y}\left(\frac{\partial U^{k}}{\partial X}+\frac{\partial V^{k}}{\partial Y}\right)dXdY\\ \;+\mathrm{Pr}\int_{\Omega }\Phi_{i}\left(\frac{\partial^{2}V^{k}}{\partial X^{2}}+\frac{\partial^{2}V^{k}}{\partial Y^{2}}\right)dXdY+Ra\mathrm{Pr}\int_{\Omega }\Phi_{i}\theta^{k}dXdY. \end{array}$$

The interpolation functions for approximating the velocity and temperature distributions can be written as: $$\begin{array}{cc} & U\approx \sum_{j=1}^{m}U_{j}\Phi_{j}\left(X,Y\right),\;V\approx \sum_{j=1}^{m}V_{j}\Phi_{j}\left(X,Y\right),\;\theta \approx \sum_{j=1}^{m}\theta_{j}\Phi_{j}\left(X,Y\right). \end{array}$$

The nonlinear residual equations for the momentum equations obtained from the Galerkin weighted residual finite-element method are: $$\begin{array}{c} R{\left(1\right)}_{i}=\sum_{j=1}^{m}U_{j}\int_{\Omega }\left[\left(\sum_{j=1}^{m}U_{j}\Phi_{j}\right)\frac{\partial \Phi_{j}}{\partial X}+\left(\sum_{j=1}^{m}V_{j}\Phi_{j}\right)\frac{\partial \Phi_{j}}{\partial Y}\right]\Phi_{i}dXdY\\ \;\;\;\;+\lambda \left[\sum_{j=1}^{m}U_{j}\int_{\Omega }\frac{\partial \Phi_{i}}{\partial X}\frac{\partial \Phi_{j}}{\partial X}dXdY+\sum_{j=1}^{m}V_{j}\int_{\Omega }\frac{\partial \Phi_{i}}{\partial X}\frac{\partial \Phi_{j}}{\partial Y}dXdY\right]\\ \;\;\;\;+\mathrm{Pr}\sum_{j=1}^{m}U_{j}\int_{\Omega }\left[\frac{\partial \Phi_{i}}{\partial X}\frac{\partial \Phi_{j}}{\partial X}+\frac{\partial \Phi_{i}}{\partial Y}\frac{\partial \Phi_{j}}{\partial Y}\right]dXdY,\\ R{\left(2\right)}_{i}=\sum_{j=1}^{m}V_{j}\int_{\Omega }\left[\left(\sum_{j=1}^{m}U_{j}\Phi_{j}\right)\frac{\partial \Phi_{j}}{\partial X}+\left(\sum_{j=1}^{m}V_{j}\Phi_{j}\right)\frac{\partial \Phi_{j}}{\partial Y}\right]\Phi_{i}dXdY\\ \;\;\;\;+\lambda \left[\sum_{j=1}^{m}U_{j}\int_{\Omega }\frac{\partial \Phi_{i}}{\partial Y}\frac{\partial \Phi_{j}}{\partial X}dXdY+\sum_{j=1}^{m}V_{j}\int_{\Omega }\frac{\partial \Phi_{i}}{\partial Y}\frac{\partial \Phi_{j}}{\partial Y}dXdY\right]\\ \;\;\;\;+\mathrm{Pr}\sum_{j=1}^{m}V_{j}\int_{\Omega }\left[\frac{\partial \Phi_{i}}{\partial X}\frac{\partial \Phi_{j}}{\partial X}+\frac{\partial \Phi_{i}}{\partial Y}\frac{\partial \Phi_{j}}{\partial Y}\right]dXdY\\ \;\;\;+Ra\mathrm{Pr}\int_{\Omega }\left(\sum_{j=1}^{m}\theta_{j}\Phi_{j}\right)\Phi_{i}dXdY, \end{array}$$
where the superscript *k* is the relative index, subscripts *i*, *j* and *m* are the residual number, the node number and the iteration number, respectively. For analyzing the nonlinear expressions in the momentum equations, a Newton–Raphson algorithm was applied. The convergence of the solution was monitored via:$$\begin{array}{c} \left|\frac{\Gamma^{m+1}-\Gamma^{m}}{\Gamma^{m+1}}\right|\leq \eta . \end{array}$$

To ensure the mesh independence of the numerical solution, various grid sizes were adopted to calculate the minimum value of the flow circulation ($\Psi_{\min}$), the interface average Nusselt number (${\bar{Nu}}_{s}$) and the average Nusselt number ($\bar{Nu}$) at N=3, Ω=250, $Ra=10^{5}$, H=0.5 and R=0.2. Table 1 shows that the difference between the solutions on grids G6 and above is insignificant. Therefore, for all similar computational problems presented in this study, the uniform grid G6 is adopted.

For validation purpose, the computed entropy generation for free convection is compared with that reported by Ilis et al. [25] as shown in Figure 3. In addition, the results for the mixed convection case agree considerably well with those of Costa and Raimundo [19] as shown in Figure 4.

## 4. Results and Discussion

In this section, the simulated streamlines, isotherms and isentropic lines for different Rayleigh numbers $(Ra=10^{3}$ and $10^{5})$, angular velocities (−1000≤Ω≤1000), heat source lengths (0.2≤H≤0.8.) and number of oscillations (1≤N≤4) are reported. Meanwhile, the other active parameters are fixed, i.e., irreversibility distribution ratio $\phi =10^{-3}$, amplitude A=0.1, dimensionless radius of the rotating cylinder R=0.2, dimensionless length of cylinder surface Θ=360 and Prandtl number Pr=4.623. The thermo-physical properties of the base liquid (water) and the solid cylinder (brickwork) are listed in Table 2.

The simulated streamline, isotherm, and isentropic lines for N=3, $Ra=10^{5}$ and H=0.5 are shown in Figure 5. For all cases, the flow is unicellular represented by a large flow cell that rotates in the clockwise direction about the solid cylinder. Indeed, the fluid flows from the hot part towards the cold part. As seen from the figure, the clockwise rotation of the cylinder increases the flow intensity, i.e., the magnitude of streamline. Nevertheless, the flow intensity decreases in the case of anti-clockwise rotation. When the rotational direction of the cylinder is similar to the flow direction in the event of natural convection, the heat transfer within the active surfaces is enhanced. At this Rayleigh number, the isotherm profiles indicate that the temperature gradient close to the vertical hot surface increases when the clockwise angular speed increases. The reverse is true when the cylinder rotates in the counter-clockwise direction. From Figure 5, the isentropic lines concentrate at the left active surface and at the top right corner of the cavity (irreversibility caused by heat transfer, HTI). Furthermore, local entropy is generated at the peaks of the horizontal wavy surfaces and near the cylinder surface as the velocity gradient is high (fluid friction irreversibility, FFI). It is noted that the counter-clockwise rotation of the cylinder generates more local entropy, particularly near the upper and lower segments of the solid cylinder surface. This is because more friction is generated when the rotational direction of the cylinder is opposite to the flow direction.

Figure 6 shows the influence of the number of oscillations *N* on the streamline, the isotherm and the isentropic lines at Ω=250, $Ra=10^{5}$ and H=0.5. The flow in the cavity is mono-cellular encompassing the rotating cylinder. It seems that the number of undulations of the surface would influence the shape of the flow cell and the distribution of isotherms adjacent to the corrugated surfaces. Besides, the streamlines and the isotherms converge at the crests and diverge at the troughs. Streamlines are highly concentrated at the crests of the undulation, showing that the fluid is accelerated (more entropy generation). At the troughs, the flow is decelerated, and the local entropy generation, $S_{\mathrm{GEN}}$ is minimal. Moreover, by increasing *N*, the temperature gradient increases while approaching the cylinder and the top part of the cold surface. Therefore, the entropy generation is intense in these areas. The large temperature and speed gradients at the top and lower sections of the cylinder tend to generate considerable amount of entropy.

The local Nusselt numbers at the heated segment and at the cylinder surface for different Ω values are presented in Figure 7. For a given Ω, the variation of local Nusselt number shows that the heat transfer rate at the lower part of the heated surface peaks at a certain position and decreases as the thermal boundary layer develops. The Nusselt number remains high at the lower part of the heated surface when Ω=0 (corresponding to the stationary cylinder or pure natural convection). At the upper segment of the hot surface, Nu is the highest at Ω=−500 (rotating cylinder in the clockwise direction). Here, we can conclude that the clockwise rotation of the cylinder (combined natural and forced convection effects) would improve the heat transfer, especially at the top section of the barrel since the flow resistance between the heated and cold surfaces is reduced (see Figure 7a). Figure 7b outlines the Nu variation over the surface of the cylinder. The counter-clockwise rotation with Ω=500 (opposite natural convection effect) gives the highest local Nusselt number as most of the heat is captured at the fluid-cylinder interface. As a result, the convective flow within the cavity is attenuated.

Figure 8a shows that the wall with two undulations are the best configuration in terms of heat augmentation. This would increase the overall convective flow within the cavity (see Figure 8b). Generally, an increasing number of waves would decrease Nu as the generated waves at the horizontal walls would block the flow inside the cavity, thus reducing the heat transfer. The first configuration is not the most suitable one as the associated convex shape would attenuate the fluid flow in the cavity (increase in velocity gradient), thus augmenting the heat gain of the cylinder and decreasing the heat transfer rate within the wavy cavity.

Figure 9a shows the variation of average Nusselt number ($\bar{Nu}$) against the dimensionless angular speed Ω and the number of oscillations *N*, at $Ra=10^{3}$ and H=0.5. For all *N* values, $\bar{Nu}$ is the lowest at the hot part when the angular speed is close to zero (nearly stationary cylinder). At this low Rayleigh number condition, for a given angular velocity, it is observed that $\bar{Nu}$ is still high when N=2. Nusselt number (${\bar{Nu}}_{i}$) is dependent on the rotational direction of the cylinder. At $Ra=10^{5}$, the mean ${\bar{Nu}}_{i}$ for the case of negative angular velocity remains higher than that of the positive counterpart. The effect of the amplitude of the angular speed in the clockwise direction on the mean ${\bar{Nu}}_{i}$ and the Bejan number is insignificant. This observation is in contrast to that observed for $Ra=10^{3}$ (see Figure 10a–c).

The contours of entropy generation do not show the relative dominance between the viscous and thermal effects. This issue can be addressed by using the Bejan number. In this context, Figure 9c shows the influence of angular velocity on the Bejan number for different wavenumbers. The Bejan number is the largest for nearly stationary cylinder (natural convection), indicating that entropy generation (irreversibility) within the cavity happens mainly through heat transfer. The increase in angular speed would decrease the Bejan number. At an angular speed of more than 500, irreversibility is mainly due to fluid friction. Furthermore, the variation of the Bejan number follows that of Nu over the cylinder surface at $Ra=10^{3}$ (Figure 9b), as the rotation of the cylinder would accelerate the fluid thus increasing the entropy generation. However, for $Ra=10^{5}$, the Bejan number is affected by the convective flow intensity, and its variation follows closely with that of ${\bar{Nu}}_{i}$ at the heated surface. By comparing Figure 9c and Figure 10c, Bejan number decays as Ra increases (higher flow speed). When heat conduction dominates, the entropy generation within the cavity is mainly due to heat transfer. However, the entropy generation is mainly driven by fluid friction in the event of convective heat transfer. Conversely, the effect of the oscillation number on the Bejan number is quite significant at higher Ra ($Ra=10^{5}$). Indeed, the Bejan number is relatively high for N=1 at $Ra=10^{3}$ and for N=4 at $Ra=10^{5}$.

From Figure 11a–c, increasing the length of the heated segment at the left vertical surface and the angular velocity Ω would increase the mean Nusselt numbers at the heater and at the cylinder surface. Meanwhile, the Bejan number is increased as well. For comparison purpose, Figure 12 shows the differences of $\bar{Nu}$, ${\bar{Nu}}_{i}$ and Bejan numbers for various *H* values at N=3 and $Ra=10^{5}$. Different Ω values are considered. The Bejan number is the maximum for the case of motionless cylinder.

## 5. Conclusions

In this work, we examined the 2D entropy production and the mixed convection inside a wavy-walled chamber containing an inner rotating cylinder and a heat source. The dimensionless governing equations and boundary conditions have been solved using the weighted residual finite element method. The effects of the problem parameters on the streamlines, isotherms, entropy lines and heat transfer rates have been shown graphically. The key findings of the study are listed below:The flow can be controlled by manipulating the angular speed of the cylinder.At low Rayleigh number, the rotational direction of the the solid cylinder does not affect the heat transfer rate, and the entropy generation is mainly driven by heat transfer for |Ω|≤500.As the Rayleigh number increases, the clockwise rotation about the solid cylinder strengthens the convective flow cell inside the wavy container. The local Nusselt number peaks at the upper segment of the heated surface.When the heat conduction dominates, the entropy generation near the wavy container is mainly driven by heat transfer. On the other hand, for convective heat transfer, entropy production is mainly driven by fluid friction.The Bejan number is high for all cases involving nearly motionless cylinder for all considered oscillation numbers.Increasing the heater length can promote the heat transfer within the wavy container. As a result, the Bejan number increases slightly.

## Figures and Tables

**Figure 1 entropy-22-00606-f001:**
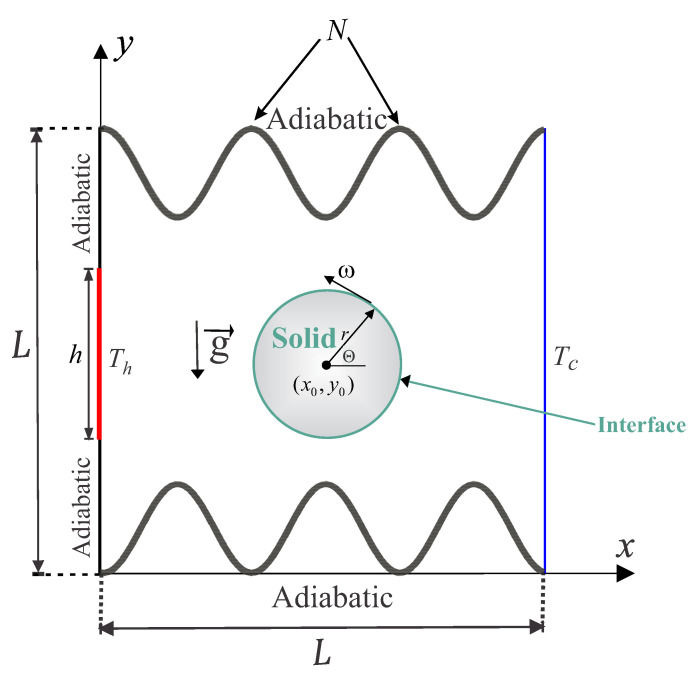
Schematic representation of the physical model and geometry.

**Figure 2 entropy-22-00606-f002:**
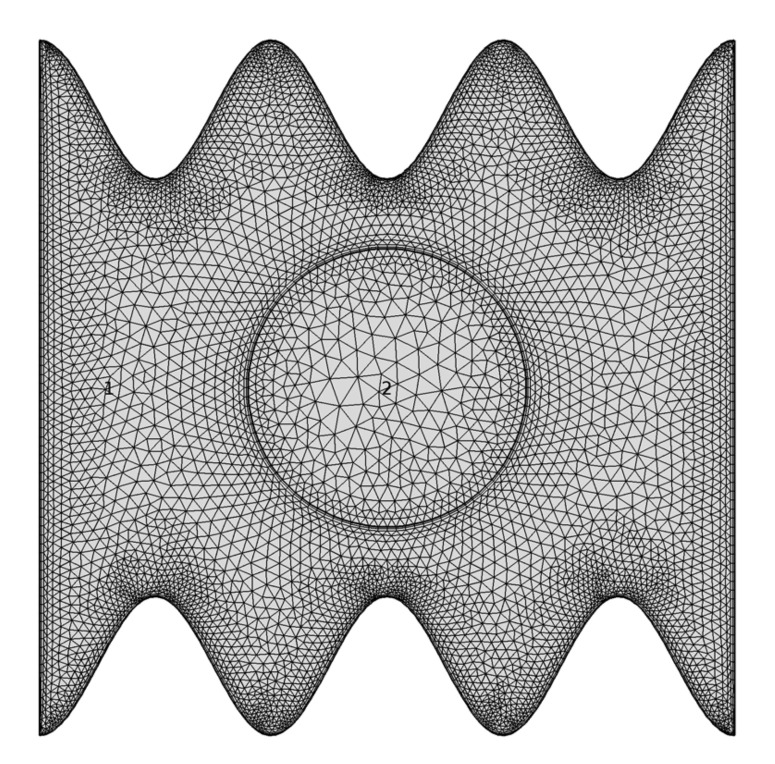
Grid-points distribution for grid size of G6.

**Figure 3 entropy-22-00606-f003:**
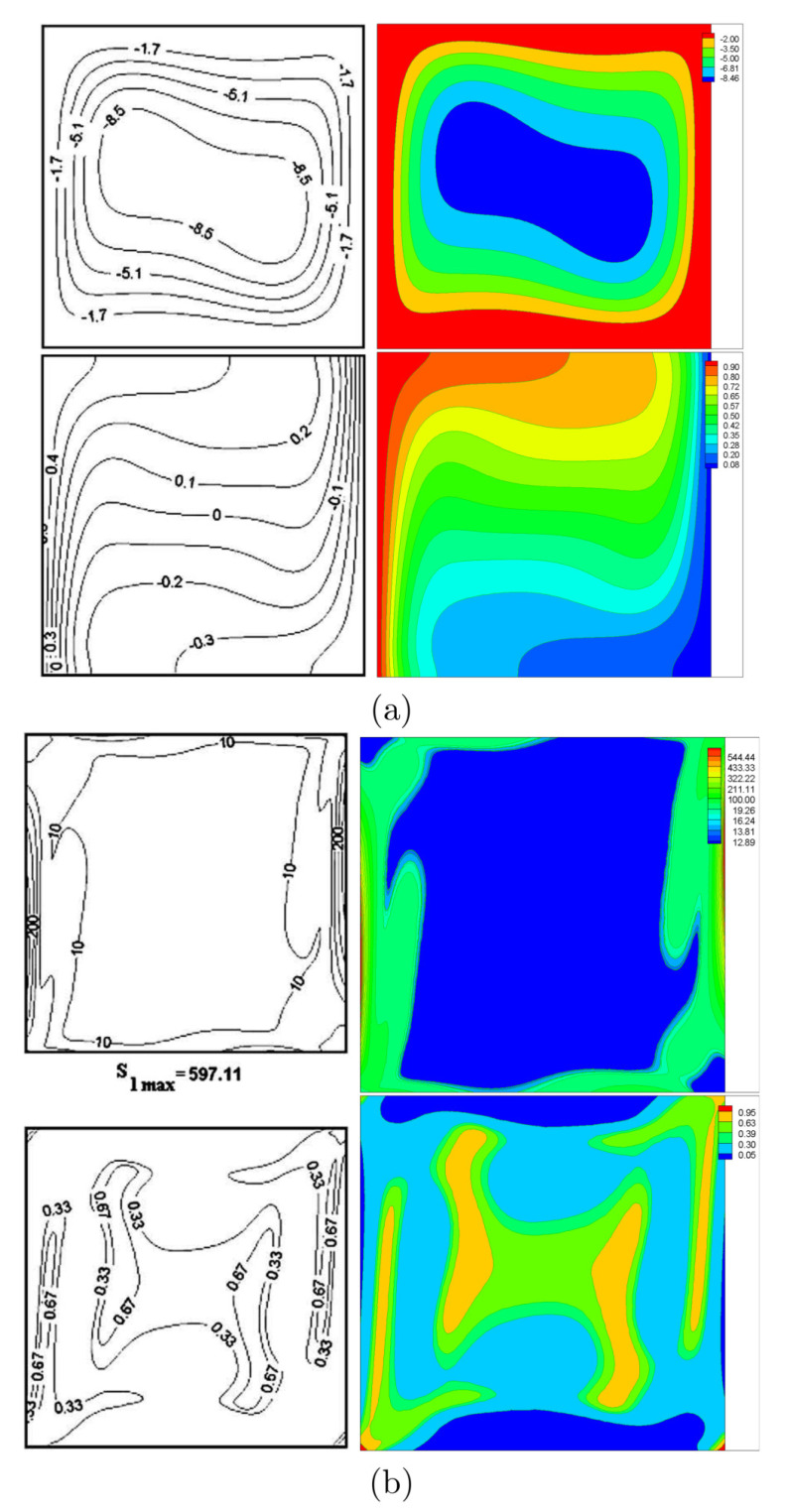
Streamlines and isotherms (**a**), global entropy generation and Bejan number (**b**), Ilis et al. [25] (left), present study (right), at Ω=0, A=0, $Ra=10^{5}$ and R=0.

**Figure 4 entropy-22-00606-f004:**
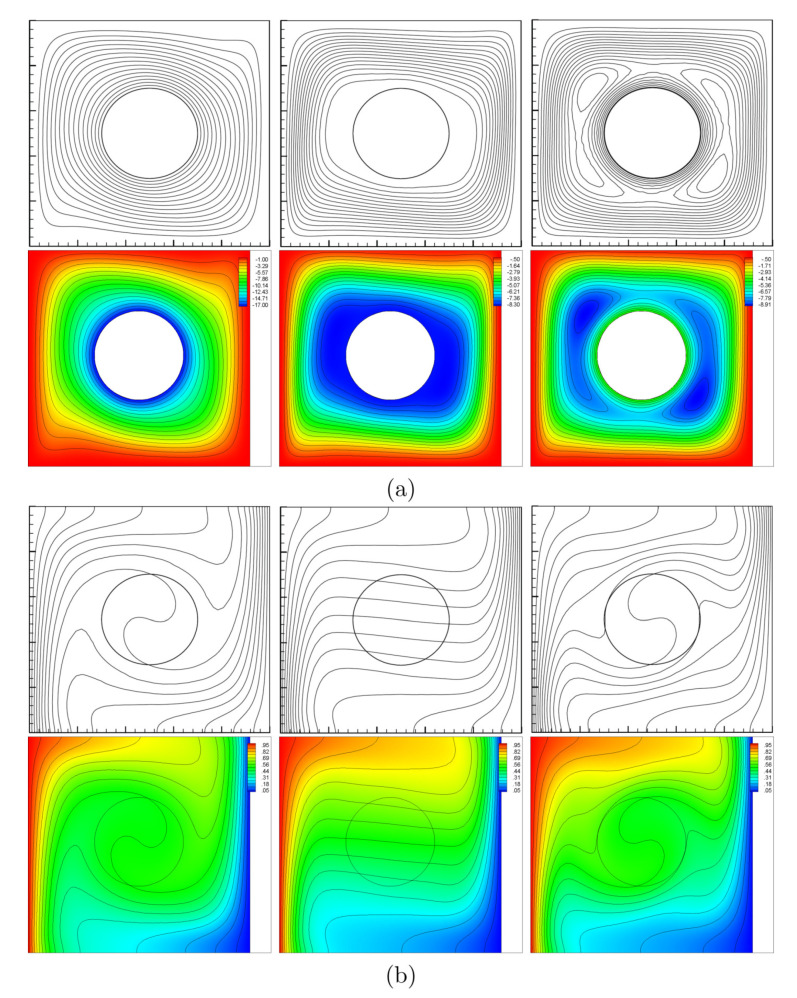
Ω=−500 (**left**), Ω=0 (**middle**), and Ω=500 (**right**) for streamlines (**a**) and isotherms (**b**); Costa and Raimundo [19] (**top**) and present study (**bottom**) at A=0, $Ra=10^{5}$, $K_{r}=1$, R=0.2 and Pr=0.7.

**Figure 5 entropy-22-00606-f005:**
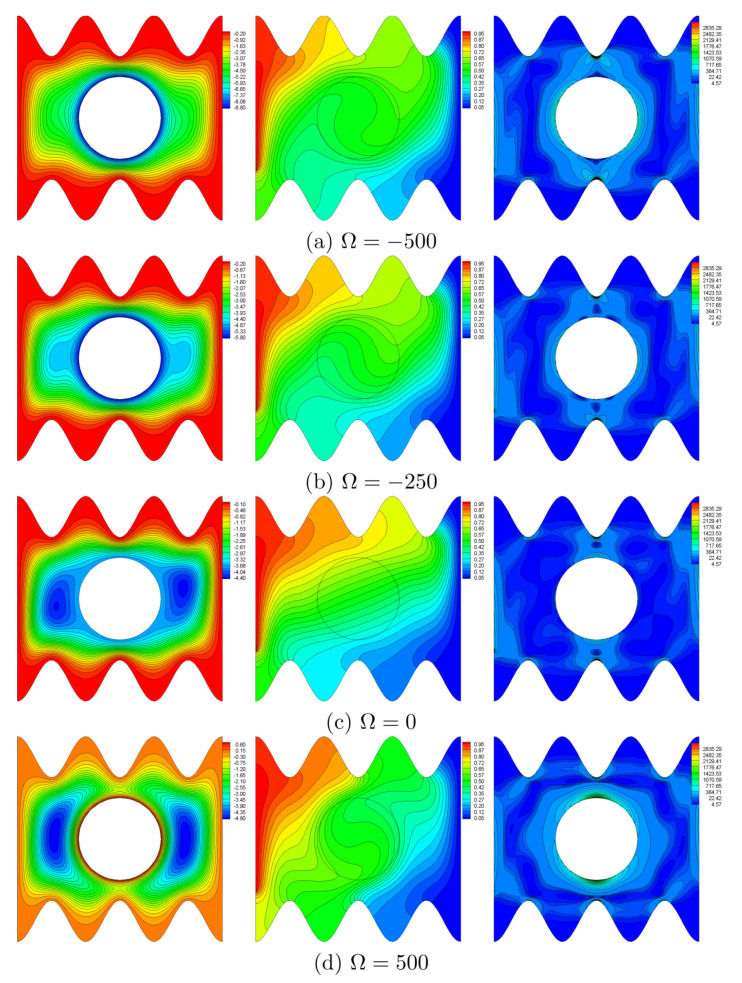
Variations of streamlines (left), isotherms (middle), and entropy lines (right) with angular rotational velocity (Ω) at N=3, $Ra=10^{5}$ and H=0.5.

**Figure 6 entropy-22-00606-f006:**
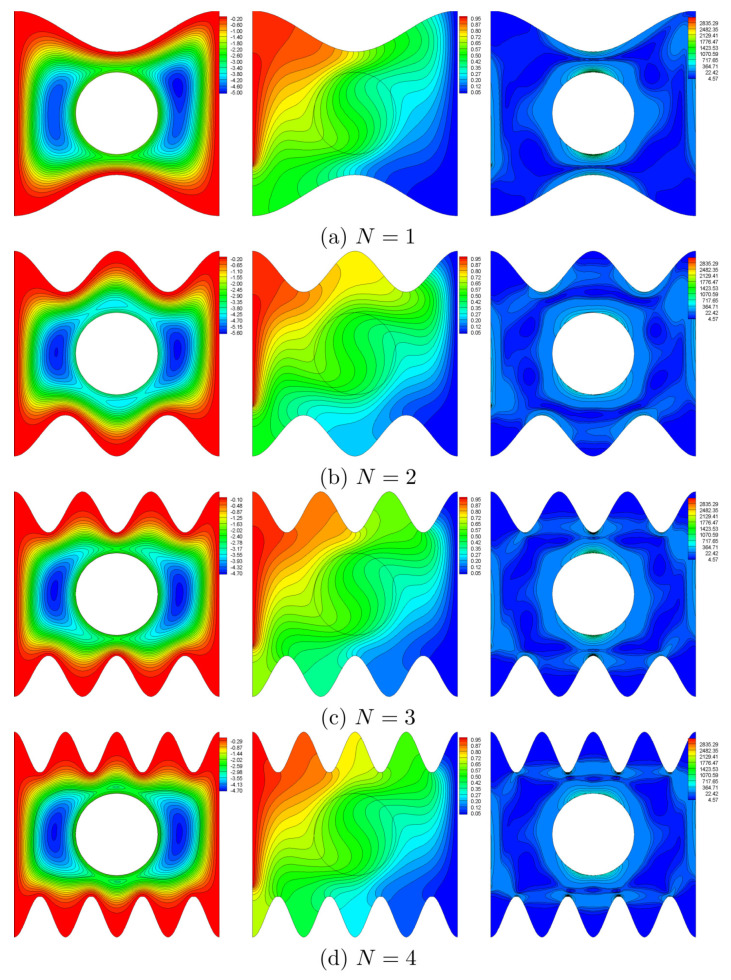
Variations of streamlines (left), isotherms (middle), and entropy lines (right) with number of oscillations (*N*) at Ω=250, $Ra=10^{5}$ and H=0.5.

**Figure 7 entropy-22-00606-f007:**
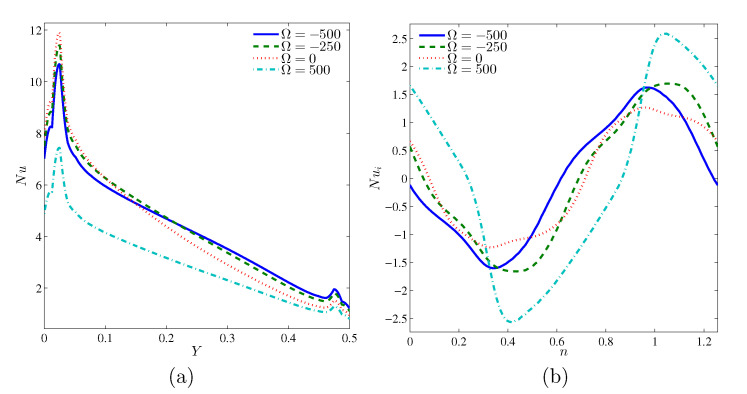
Variations of local Nusselt numbers at the interface with *Y* (**a**) and *n* (**b**) for various Ω at N=3, $Ra=10^{5}$ and H=0.5.

**Figure 8 entropy-22-00606-f008:**
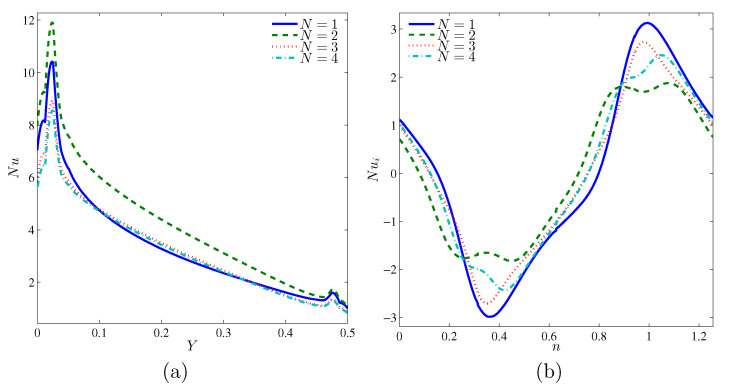
Variations of local Nusselt number at the interfaces with *Y* (**a**) and *n* (**b**) for various *N* at Ω=250, $Ra=10^{5}$ and H=0.5.

**Figure 9 entropy-22-00606-f009:**
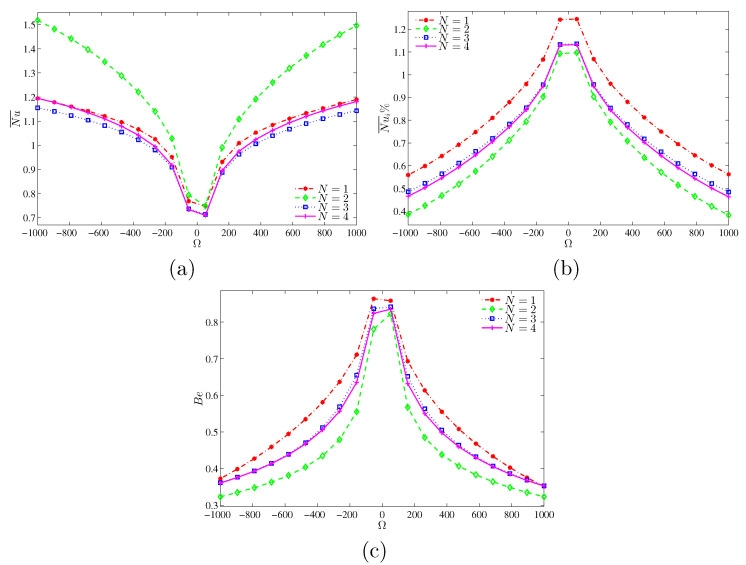
Variations of (**a**) average Nusselt number, (**b**) interface average Nusselt number and (**c**) interface Bejan number with Ω for various *N* at $Ra=10^{3}$ and H=0.5.

**Figure 10 entropy-22-00606-f010:**
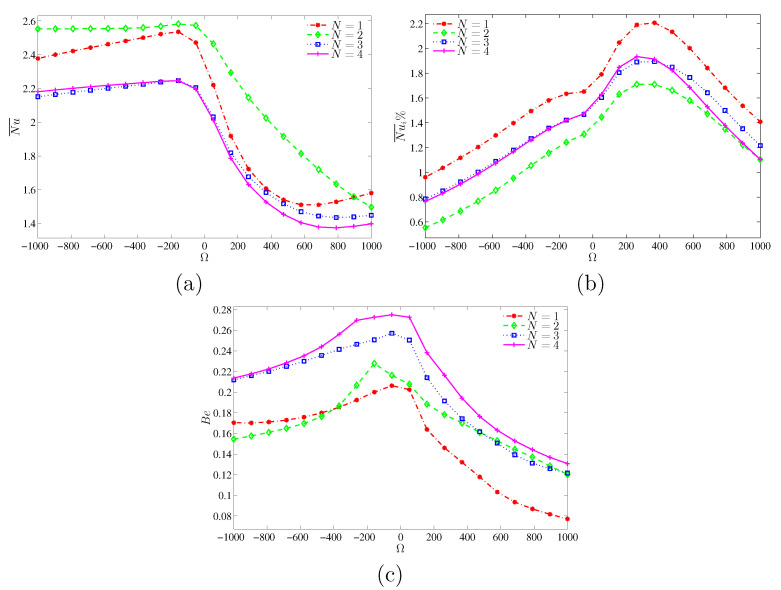
Variations of (**a**) average Nusselt number, (**b**) interface average Nusselt number and (**c**) Bejan number interface with Ω for various *N* at $Ra=10^{5}$ and H=0.5.

**Figure 11 entropy-22-00606-f011:**
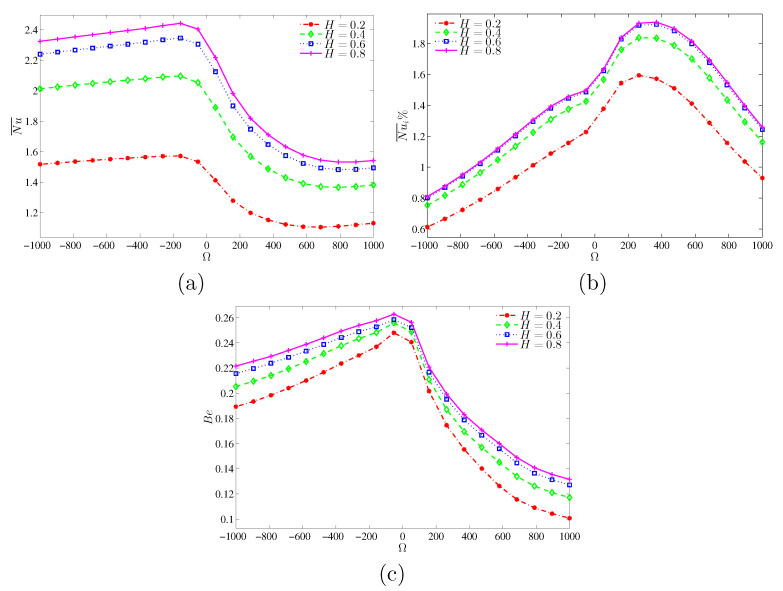
Variations of (**a**) average Nusselt number, (**b**) interface average Nusselt number and (**c**) interface Bejan number with Ω for various *H* at N=3 and $Ra=10^{5}$.

**Figure 12 entropy-22-00606-f012:**
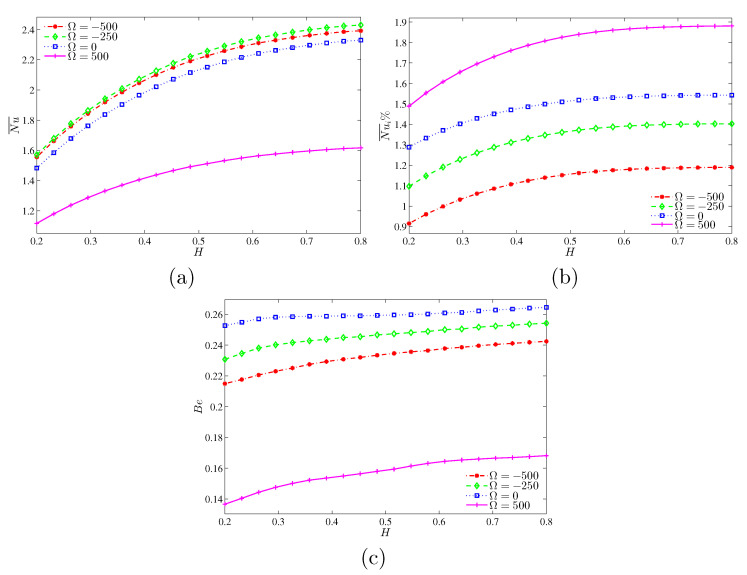
Variations of (**a**) average Nusselt number, (**b**) interface average Nusselt number and (**c**) interface Bejan number with *H* for various Ω at N=3 and $Ra=10^{5}$.

**Table 1 entropy-22-00606-t001:** Grid independent test for $\Psi_{\min}$, ${\bar{Nu}}_{s}$ and $\bar{Nu}$ at Ω=250, H=0.5, N=3, $Ra=10^{5}$ and R=0.2.

Grid Size	Number of Elements	$\Psi_{\min}$	${\bar{\mathrm{Nu}}}_{s}$	$\bar{\mathrm{Nu}}$
G1	2971	−4.6663	1.4835	3.2209
G2	3403	−4.703	1.4927	3.2547
G3	3909	−4.7297	1.4946	3.2587
G4	4810	−4.7395	1.4982	3.2832
G5	11794	−4.7682	1.5015	3.3818
**G6**	27151	−4.7762	1.509	3.3833
G7	32745	−4.7793	1.5088	3.3836

**Table 2 entropy-22-00606-t002:** Thermo-physical properties of the base liquid (water) and solid cylinder (brickwork).

Physical Properties	Fluid Phase (Water)	Solid Cylinder (Brickwork)
$k\;({\mathrm{Wm}}^{-1}K^{-1})$	0.628	0.76
$\rho \;(\mathrm{kg}/m^{3})$	993	1700
$C_{p}\;\left(J/\mathrm{kgK}\right)$	4178	800

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
