# Peer review of "Entropy Generation and Mixed Convection Flow Inside a Wavy-Walled Enclosure Containing a Rotating Solid Cylinder and a Heat Source"

_entropy, 2020, doi:10.3390/e22060606_

Round 1

Reviewer 1 Report

The authors have presented a numerical study on entropy generation and mixed convection inside a wavy-walled enclosure containing a rotating solid cylinder and a heat source. The paper is interesting and has potential to be a journal paper. But it is written poorly, a lot of work is needed to improve the overall quality and readability of the paper. The paper in its current state cannot be recommended for publication. The following should be addressed in the revised paper.    

(1) Line 1. Change "The current study analyzed the entropy production plus the mixed convection inside a wavy chamber holding an interior rotating cylinder." to "The current study analyzed the entropy production and the mixed convection inside a wavy-walled chamber holding an interior rotating cylinder and a heat source."

(2) Line 2. Change "The heat source by finite-length h does placed toward the middle of the left vertical surface and sustained by a fixed hot temperature Th" to "The heat source is of finite-length h and it is placed close to the middle of the left vertical surface and maintained at a fixed hot temperature Th"

(3) Line 3. Change "right vertical surface remains on a fixed cold temperature Tc" to "right vertical surface remains at a fixed cold temperature Tc"

(4) Line 14. Change "nearly stationary cylinder" to "nearly stationary inner cylinder"

(5) Line 18. Please split, rewrite and improve the first sentence. It is poorly written and confusing.

(6) Line 24. Change "number" to "interest"

(7) Line 89, Change "Toward the knowledge of the authors, the case concerning the entropy generation and mixed convection inside cavity with horizontal wavy walls having a solid rotating circular cylinder should not be explained yet." to "To the knowledge of the authors, the case concerning the entropy generation and mixed convection inside a cavity with horizontal wavy walls having a solid rotating circular cylinder has not been studied yet."

***Please Note this reviewer did not correct the whole paper for this kind of mistakes. The authors should check the whole paper and look for typos and grammatical mistakes.  

(8) The introduction can be improved by providing pictures of practical engineering applications relevant to the problem addressed in the paper. This will make the paper interesting and provide the reason for the motivation behind this work.  

(9) The geometry considered in this study should already must be explained in the introduction section in the beginning of the paper. This will help the reader follow the paper better.

(10) There are a lot of symbols and abbreviations in this paper. The authors should provide a nomenclature with all the symbols and abbreviations used in the paper. It will be useful for the readers and reviewers.

(11) Line 157, Change "Current part performs the numerical outcomes of streamlines, isotherms and isentropic lines157 toward varying Rayleigh numbers …" to "In this section, results from the numerical simulations are discussed using streamlines, isotherms and isentropic lines for varying Rayleigh numbers …"

(12) Line 159, Change "others" to "other". Did anybody proof read this paper. There are so many typos and grammatical mistakes. The authors should properly proof read the paper before submitting again.

(13) Line 250, Change "expressed" to "obtained"

(14) The authors should consider changing the title from "Rules of rotating solid cylinder and heat source on mixed convection and entropy generation inside wavy-walled enclosure" to "Entropy generation and mixed convection inside a wavy-walled enclosure containing a rotating solid cylinder and a heat source"

(15) Does the geometry considered in this paper correspond to a particular practical application or an experimental study. If so mention it and discuss in the last paragraph of introduction. Also include the references in the introduction and also in section 2.

Author Response

First of all, we thank the respected referees for their constructive comments which clearly enhanced the quality of the manuscript. Our replies to the comments are given below:

Reviewer #1
Comment 1. Line 1. Change “The current study analyzed the entropy production plus the mixed
convection inside a wavy chamber holding an interior rotating cylinder.” to “The
current study analyzed the entropy production and the mixed convection inside a
wavy-walled chamber holding an interior rotating cylinder and a heat source.”
Reply: We have revised the the above mentioned sentence according to the respected
reviewer as:
“The current study analyzed the entropy production and the mixed convection inside
a wavy-walled chamber holding an interior rotating cylinder and a heat source”.
Comment 2. Line 2. Change “The heat source by finite-length h does placed toward the middle
of the left vertical surface and sustained by a fixed hot temperature Th” to “The heat
source is of nite-length h and it is placed close to the middle of the left vertical
surface and maintained at a fixed hot temperature Th”.
Reply: We have revised the the above mentioned sentence according to the respected
reviewer as:
“The heat source is of finite-length h, and it is placed close to the middle of the left
vertical surface and maintained at a fixed hot temperature Th”.
Comment 3. Line 3. Change “right vertical surface remains on a fixed cold temperature Tc” to
“right vertical surface remains at a fixed cold temperature Tc”.
Reply: We have revised the the above mentioned sentence according to the respected
reviewer as:
“right vertical surface remains at a fixed cold temperature Tc.
Comment 4. Line 14. Change “nearly stationary cylinder” to “nearly stationary inner cylinder”.
Reply: Done.
Comment 5. Line 18. Please split, rewrite and improve the first sentence. It is poorly written and
confusing.
Reply: We are grateful for the Reviewer’s notes about the above mentioned sentences
as we have revised to be:
“Natural convection and mixed convection mechanisms concerning enclosed cavities
with various shapes have taken a great interest and possible applications in
engineering as reported in many early works”.
Comment 6. Line 24. Change “number” to “interest”.
Reply: Done.
Comment 7. Line 89, Change “Toward the knowledge of the authors, the case concerning the
entropy generation and mixed convection inside cavity with horizontal wavy walls
having a solid rotating circular cylinder should not be explained yet.” to “To the
knowledge of the authors, the case concerning the entropy generation and mixed
convection inside a cavity with horizontal wavy walls having a solid rotating circular
cylinder has not been studied yet.”
**Please Note this reviewer did not correct the whole paper for this kind of mistakes.
The authors should check the whole paper and look for typos and grammatical
mistakes.
Reply: We have revised the the above mentioned sentence according to the respected
reviewer as:
“To the knowledge of the authors, the case concerning the entropy generation and
mixed convection inside a cavity with horizontal wavy walls having a solid rotating
circular cylinder has not been studied yet...
Comment 8. The introduction can be improved by providing pictures of practical engineering
applications relevant to the problem addressed in the paper. This will make the
paper interesting and provide the reason for the motivation behind this work.
Reply: We have redrafted the last paragraph of the Introduction section of the revised
manuscript to include some of the practical engineering applications related to the
present problem. Thank you for pointing this out.
Comment 9. The geometry considered in this study should already must be explained in the
introduction section in the beginning of the paper. This will help the reader follow
the paper better.
Reply: Thank you for pointing out this aspect. We have addressed this issue in the first
paragraph of the introduction section, and we have also explained the considered
geometry in the last paragraph of the introduction. Please see the highlighted parts
in the revised manuscript.
Comment 10. There are a lot of symbols and abbreviations in this paper. The authors should provide
a nomenclature with all the symbols and abbreviations used in the paper. It
will be useful for the readers and reviewers.
Reply: We have included a nomenclature section to the revised version.
Comment 11. Line 157, Change “Current part performs the numerical outcomes of streamlines,
isotherms and isentropic lines157 toward varying Rayleigh numbers ...” to “In this
section, results from the numerical simulations are discussed using streamlines,
isotherms and isentropic lines for varying Rayleigh numbers ...”
Reply: We have revised revised the above mentioned sentence to be: “In this section,
results from the numerical simulations are discussed using streamlines, isotherms
and isentropic lines for varying Rayleigh numbers...”.
Comment 12. Line 159, Change “others” to “other”. Did anybody proof read this paper. There are
so many typos and grammatical mistakes. The authors should properly proof read
the paper before submitting again.
Reply: The authors truly apologize for the accidental mistakes in the writing of the
manuscript. This because we have made several revisions on the manuscript to avoid
the similarity problems. We have carefully proofread the revised manuscript and
corrected all the errors in English grammar and typos.
2
Comment 13. Line 250, Change “expressed” to “obtained”.
Reply: Done.
Comment 14. The authors should consider changing the title from ”Rules of rotating solid cylinder
and heat source on mixed convection and entropy generation inside wavy-walled
enclosure” to ”Entropy generation and mixed convection inside a wavy-walled enclosure
containing a rotating solid cylinder and a heat source”.
Reply: We agree with the suggestion and the new title now reads as:
“Entropy generation and mixed convection flow inside a wavy-walled enclosure
containing a rotating solid cylinder and a heat source”.
Comment 15. Does the geometry considered in this paper correspond to a particular practical application
or an experimental study. If so mention it and discuss in the last paragraph
of introduction. Also include the references in the introduction and also in section
2.
Reply: This present configuration can be seen in many practical industrial and engineering
applications. We have redrafted the last paragraph of the Introduction section
of the revised manuscript to include some of the practical engineering applications
related to the present problem with references.

Thank you for your support.

Sincerely yours,

Reviewer 2 Report

Please, see the enclosed file.

Author Response

First of all, we thank the respected referees for their constructive comments which clearly enhanced the quality of the manuscript. Our replies to the comments are given below:

Reviewer #2
Comment 1. Please, explain the used definition for the local Nusselt number at cylinder surface
presented in Eq. (19).
Reply: We have followed the definition of the local Nusselt number that used in Costa
and Raimundo [18] as:

Comment 2. What value of the irreversibility parameter was used for analysis and why?
Reply:We have included the value of the irreversibility distribution ratio which is 10?3
(on line 163) following most of the previous published papers.
Comment 3. What software was used for the mesh generation?
Reply: We used Matlab along with COMSOL 5.2 with MATLAB for the mesh genera-
tion.
Comment 4. Please, explain the behavior of local Nusselt number at the local heater presented
in Figs. 7(a) and 8(a). I mean the formation of local extrema near the ends of the
heater.
Reply: The local Nusselt records its maximum at the lower edge of the heater because
in this region we notice a large concentration of isotherms (please see isotherms in
Figs. 5 and 6) corresponding to the contact of the arrived cold fluid from the right
wall with the lower end of the heater, which will give high-temperature gradients in
this area, and consequently, leads to the maximum local heat transfer rate.
Comment 5. What is the reason for the non-monotonic behavior of the average Nusselt numbers
with undulation number presented in Figs. 9 and 10?
Reply: Generally mean Nusselt numbers decrease as the undulations number increases.
This is because the presence of undulations on the horizontal walls block the fluid
flow within the cavity and then reduces the heat transfer rate. The reason why the
first configuration (N = 1) is not the best compared with the second one (N = 2)
is purely geometric as the convex shape of the first configuration leads to narrows
the fluid flow inside the cavity and increases the velocity gradient on the cylinder
surface, resulting in an augmentation in heat gain by the cylinder and consequently
a decreasing in heat transfer rate inside the enclosure.
Comment 6. Introduction part can be improved with additional papers on entropy generation in
wavy enclosures. You can read the following papers:
Entropy 18(9) (2016) doi:10.3390/e18010009.
Journal of Molecular Liquids 263 (2018) 510525.
Reply: We have updated the literature survey to contain the mentioned articles.
Comment 7. There are some typos within the text. Please, reread the text and correct all typos.
Reply: We have carefully proofread the revised manuscript and corrected the punctuation in the revised manuscript.

Thank you for your support.

Sincerely yours,

Round 2

Reviewer 1 Report

The authors have implemented all the corrections suggested in the review report. As a result the quality of the paper is much improved. I can recommend publication of the paper in the journal after checking the English.  

Author Response

We have carefully proofread the revised manuscript and corrected the punctuation in the the revised manuscript.